



# TCCON and NDACC $X_{CO}$ measurements: difference, discussion and application

Minqiang Zhou[1], Bavo Langerock[1], Corinne Vigouroux[1], Mahesh Kumar Sha[1], Christian Hermans[1], Jean-Marc Metzger[2], Huilin Chen[3], Michel Ramonet[4], Rigel Kivi[5], Pauli Heikkinen[5], Dan Smale[6], David F. Pollard[6], Nicholas Jones[7], Voltaire A. Velazco[7], Omaira E. García[8], Matthias Schneider[9], Mathias Palm[10], Thorsten Warneke[10], and Martine De Mazière[1]

[1]Royal Belgian Institute for Space Aeronomy (BIRA-IASB), Brussels, Belgium
[2]UMS 3365 – OSU Réunion, Université de La Réunion, Saint-Denis, Réunion, France
[3]Centre for Isotope Research (CIO), Energy and Sustainability Research Institute Groningen (ESRIG), University of Groningen (RUG), Groningen, the Netherlands
[4]Laboratoire des Sciences du Climat et de l'Environnement (LSCE/IPSL), UMR CEA-CNRS-UVSQ, Gif-sur-Yvette, France
[5]Finnish Meteorological Institute (FMI), Space and Earth Observation Centre, Sodankylä, Finland
[6]National Institute of Water and Atmospheric Research (NIWA), Lauder, New Zealand
[7]Centre for Atmospheric Chemistry, University of Wollongong, Wollongong, Australia
[8]Izaña Atmospheric Research Centre (IARC), Meteorological State Agency of Spain (AEMET), Santa Cruz de Tenerife, Spain
[9]Karlsruhe Institute of Technology (KIT), Institute of Meteorology and Climate Research, Karlsruhe, Germany
[10]Institute of Environmental Physics, University of Bremen, Bremen, Germany

**Correspondence:** Minqiang Zhou (minqiang.zhou@aeronomie.be)

**Abstract.** Column-averaged dry-air mole fraction of CO ($X_{CO}$) measurements are obtained from two ground-based Fourier transform infrared (FTIR) spectrometers networks: the Total Carbon Column Observing Network (TCCON) and the Network for the Detection of Atmospheric Composition Change (NDACC). In this study, the differences between the TCCON and NDACC $X_{CO}$ measurements are investigated and discussed based on six NDACC/TCCON sites using data over the period 2007-2017. The NDACC $X_{CO}$ measurements are about 5.5% larger than the TCCON data at Ny-Ålesund, Bremen, and Izaña (Northern Hemisphere), and about 0.3% larger than the TCCON data at St Denis, Wollongong and Lauder (Southern Hemisphere). The hemispheric dependence of the bias is mainly attributed to their smoothing errors. The systematic smoothing error of the TCCON $X_{CO}$ data varies in the range between 0.2% (Bremen) and 7.9% (Lauder), and the random smoothing error in the range between 2.0% and 3.6%. The systematic smoothing error of NDACC data is between 0.1% and 0.8%, and the random smoothing error of NDACC data is about 0.3%. For TCCON data, the smoothing error can be significant in that it is much higher than the reported uncertainty for TCCON $X_{CO}$. To reduce the influence from the a priori profiles and different vertical sensitivities, the scaled NDACC a priori profiles are used as the common a priori profiles for comparing TCCON and NDACC retrievals. As a result, the biases between TCCON and NDACC $X_{CO}$ measurements become more consistent (5.6-8.5%) with a mean value of 6.8% at these sites. To understand the remaining bias, regular AirCore measurements at Orleans and Sodankylä are compared to co-located TCCON measurements. It is found that TCCON $X_{CO}$ measurements are 6.0 ± 1.9% and 6.9 ± 2.5% smaller than the AirCore measurements at Orleans and Sodankylä respectively, indicating that the scaling factor of



TCCON $X_{CO}$ data should be around 1.0000 instead of 1.0672. Further investigations should be carried out in the TCCON community to determine the correct scaling factor to be applied to the TCCON $X_{CO}$ data. This paper also demonstrates that the smoothing error must be taken into account when comparing FTIR $X_{CO}$ data, and especially TCCON $X_{CO}$ data, with model or satellite data.

## 5  1  Introduction

Carbon monoxide (CO) is a trace gas in the Earth's atmosphere, with a typical mole fraction of 50 - 80 ppb (parts per billion) at clean-air sites. Atmospheric CO is released by incomplete combustion, mainly coming from anthropogenic emissions (Granier et al., 2011), biomass burnings (van der Werf et al., 2010). There is also small qualities of CO in the mesosphere generated by the photolysis of carbon dioxide (Garcia et al., 2014). The lifetime of CO is about two months in the troposphere (Pfister
et al., 2004), and on the order of several months in the stratosphere (Hoor et al., 2004). CO is often used as a tracer to study the long-distance transport of biomass burnings (Duflot et al., 2010), wildfires (Turquety et al., 2009) and anthropogenic emissions (Ojha et al., 2016). The major sink of CO in the atmosphere is the reaction with hydroxyl radicals (OH) (Spivakovsky et al., 2000). Therefore, CO plays an important role in atmospheric chemistry and thus affecting the atmospheric oxidizing capacity. CO concentration is associated with many tropospheric polluting gases, e.g., tropospheric ozone and urban smog (Aschi and
Largo, 2003), and it also has a strong impact on the carbon and methane cycles (Rasmussen and Khalil, 1981).

Global CO total columns are measured by space-based satellite instruments, e.g. the measurement of pollution in the troposphere (MOPITT), the scanning imaging absorption spectrometer for atmospheric cartography (SCIAMACHY), the infrared atmospheric sounding interferometer (IASI) and the more recently tropospheric monitoring instrument (TROPOMI) (Deeter et al., 2017; Borsdorff et al., 2016; George et al., 2009; Borsdorff et al., 2018). Satellite measurements are applied to study
the long-term trend of CO (Worden et al., 2013), to understand the regional pollution (Dekker et al., 2019) and are assimilated into the atmospheric chemistry model (Klonecki et al., 2012; Mizzi et al., 2016). To better understand the uncertainties of the satellite CO observations and the model simulations, they need to be validated by other measurements. Ground-based Fourier transform infrared (FTIR) spectrometers record the direct solar radiation and observe the total column of CO with high accuracy and precision. In addition, the ground-based FTIR CO measurements are stable over a long-time period, so that they can
be used to validate the satellite CO observations (Dils et al., 2006; Borsdorff et al., 2016, 2018) and model simulations (Eskes et al., 2015). Nowadays, there are two well-known global ground-based FTIR networks providing total column-averaged dry air mole fraction of CO ($X_{CO}$) measurements: the Total Carbon Column Observing Network (TCCON) (Wunch et al., 2011) and the Network for the Detection of Atmospheric Composition Change (NDACC) (De Mazière et al., 2018).

TCCON and NDACC $X_{CO}$ measurements are sometimes combined together to validate satellite observations or model
simulations, and it is noticed that the smoothing error of TCCON and NDACC $X_{CO}$ measurements are not always taken into account when comparing with satellite observations, e.g. SCIAMACHY (Borsdorff et al., 2016; Hochstaffl et al., 2018) and TROPOMI (Borsdorff et al., 2018) because it is considered to have a negligible impact. By using both TCCON and NDACC $X_{CO}$ data to validate the SCIAMACHY observations, Borsdorff et al. (2016) found that NDACC $X_{CO}$ data is 3.8



ppb larger than TCCON measurements. Despite of the similar measurement technique, there are differences between TCCON and NDACC $X_{CO}$ products because the observed spectra, retrieval algorithms and data corrections are different. To understand why there is a systematic bias between the TCCON and NDACC $X_{CO}$ measurements, a case study was carried out by Kiel et al. (2016) using TCCON and NDACC measurements at Karlsruhe during 2010-2014. They found that NDACC $X_{CO}$ is

$4.47 \pm 0.17$ ($1\sigma$) ppb larger than the TCCON data, and the difference between the TCCON and NDACC $X_{CO}$ measurements mainly comes from the airmass independent (scaling) correction of the TCCON data, and partly from the airmass dependent correction, the spectroscopic parameters and a priori profiles.

In this study, the comparison between the TCCON and NDACC $X_{CO}$ measurements is extended to six sites (Ny-Ålesund, Bremen, Izaña, St Denis, Wollongong and Lauder) during the time period of 2007-2017. This work aims at understanding

(1) whether the bias between TCCON and NDACC $X_{CO}$ measurements is consistent at these sites, (2) whether the smoothing uncertainties of TCCON and NDACC $X_{CO}$ measurements can be ignored when comparing against each other or other datasets, and (3) whether the scaling factor of TCCON $X_{CO}$ data is correct. This paper is organized as follows. Section 2 lists the FTIR sites used in this study and describes the main characteristics of the TCCON and NDACC $X_{CO}$ measurements. Direct comparisons between TCCON and NDACC $X_{CO}$ measurements are carried out in the next section. In Section 4, the differences

between TCCON and NDACC $X_{CO}$ measurements are investigated as to their a priori profiles and averaging kernels. The smoothing errors of TCCON and NDACC $X_{CO}$ measurements are estimated. The TCCON $X_{CO}$ measurements are compared with AirCore measurements at Sodankylä and Orleans. Section 5 shows an example of using TCCON and NDACC $X_{CO}$ measurements together in a comparison with a model simulation. Conclusions are drawn in Section 6.

## 2   FTIR measurements

The ground-based FTIR measurement system is composed of an automatic weather station, a sun tracker and a FTIR instrument. The locations of the FTIR sites used in this study and time coverages of the TCCON and NDACC $X_{CO}$ measurements are listed in Table 1. All these sites use a Bruker IFS 120/125HR instrument to record near infrared (NIR) spectra for TCCON measurements and mid infrared (MIR) spectra for NDACC measurements. The main characteristics of TCCON and NDACC $X_{CO}$ measurements are described below.

### 2.1   TCCON


TCCON uses the GGG2014 code that applies a profile scaling to retrieve CO and $O_2$ total columns simultaneously (Wunch et al., 2015). The spectral resolution of the NIR spectrum is $0.02~cm^{-1}$. The retrieval windows of CO are 4208.7-4257.3 $cm^{-1}$ and 4262.0-4318.8 $cm^{-1}$. The interfering species are $CH_4$, $H_2O$ and HDO. The retrieval window of $O_2$ is 7765.0 - 8005.0 $cm^{-1}$, with interfering absorptions from $H_2O$, HF, $CO_2$ and solar lines. The spectroscopy is the ATM linelist maintained at

JPL, NASA (Toon, 2014). Since the $O_2$ volume mixing ratio (VMR) of 0.2095 is constant in the atmosphere, TCCON uses the $O_2$ total column ($TC_{O_2}$) to calculate the total column of the dry air ($TC_{dry,air} = TC_{O_2}/0.2095$), and then to calculate the $X_{CO}$ as the ratio between the retrieved CO total column and the total column of the dry air ($X_{CO} = 0.2095 \times \frac{TC_{CO,r}}{TC_{O_2,r}}$).



**Table 1.** The coordinates, responsible institute and time coverage of measurements at six sites used in this study.

| Site | Latitude | Longitude | Altitude (km a.s.l) | Research group | Time coverage (TCCON/NDACC) | Instrument |
|------|----------|-----------|---------------------|----------------|-----------------------------|------------|
| Ny-Ålesund | 78.9°N | 11.9°E | 0.02 | U. of Bremen | 2007-2017/2007-2017 | Bruker 120HR |
| Bremen | 53.1°N | 8.8°E | 0.03 | U. of Bremen | 2009-2017/2007-2016 | Bruker 125HR |
| Izaña | 28.3°N | 16.5°W | 2.37 | AEMET & KIT | 2007-2017/2007-2017 | Bruker 125HR |
| St Denis (Reunion Island) | 21.0°S | 55.4°E | 0.08 | BIRA-IASB | 2011-2017/2011-2015 | Bruker 125HR |
| Wollongong | 34.4°S | 150.9°E | 0.03 | U. of Wollongong | 2008-2017/2008-2017 | Bruker 125HR |
| Lauder | 45.0°S | 169.7°E | 0.37 | NIWA | 2010-2017/2007-2017 | Bruker 120/5HR |

Furthermore, TCCON $X_{CO}$ data have been indirectly validated by several aircraft and AirCore measurements, and the publicly available TCCON $X_{CO}$ data have been corrected with a scaling factor ($\alpha$) and an airmass dependent factor ($\beta$) (Wunch et al., 2015)

$$X_{CO} = 0.2095 \times \frac{TC_{CO,r}}{TC_{O_2,r}} \times \frac{1}{\alpha \cdot [1 + \beta \times SBF(\theta)]}, \tag{1}$$

where $\alpha = 1.0672$ and $\beta = -0.0483$, $\theta$ is the solar zenith angle (SZA) and the $SBF(\theta)$ depends on the probed airmass through the SZA ($SBF(\theta) = [(\theta + 13)/(90 + 13)]^3 - [(45 + 13)/(90 + 13)]^3$).

According to Figure 10 in Wunch et al. (2015), the random uncertainty of TCCON $X_{CO}$ data is below 3.5% and decreases with increasing SZA. The largest source is the uncertainty of the observer-sun Doppler stretch (osds) due to a solar tracker pointing uncertainty. The shear misalignment, continuum curvature and a priori profile shape are the other leading sources of uncertainty, and they are all about 1.0%. In this study, it is assumed that the mean random uncertainty of TCCON $X_{CO}$ measurement is 3.0% as a upper limitation. Since TCCON data have been scaled to the WMO standard, the systematic uncertainty of TCCON $X_{CO}$ data is eliminated and it is assumed to be zero. Note that the systematic smoothing error has not been removed in pubic TCCON data, because the Aircraft or AirCore profiles which are used to calibrate the TCCON $X_{CO}$ measurements have been first smoothed with TCCON data (Wunch et al., 2010).

## 2.2 NDACC

NDACC uses either the SFIT4 (Pougatchev et al., 1995) or the PROFFIT9 code (Hase et al., 2004) to retrieve CO vertical profiles. The retrieval windows for CO are 2057.70-2058.00 $cm^{-1}$, 2069.56-2069.76 $cm^{-1}$ and 2157.50-2159.15 $cm^{-1}$. The spectral resolution of the MIR spectrum is about 0.0035 - 0.007 $cm^{-1}$. The interfering species are $O_3$, $CO_2$, OCS, $N_2O$ and $H_2O$. The spectroscopy is HITRAN 2008 (Rothman et al., 2009). Since the $O_2$ total column is not available from the NDACC spectrum and the weak $N_2$ (a potential alternative) signal in the NDACC region leads to a large scatter, the total column of the dry air is computed from the surface pressure ($P_s$) recorded at a local automatic weather station and the National Centers for





Environmental Prediction (NCEP) reanalysis $H_2O$ total column ($TC_{H_2O}$)

$$X_{CO} = \frac{TC_{CO,r}}{TC_{dry,air}} = \frac{TC_{CO,r}}{P_s/(gm_{air}^{dry}) - TC_{H_2O}(m_{H_2O}/m_{air}^{dry})}, \tag{2}$$

where $g$ is the column-averaged gravity acceleration, $m_{H_2O}$ and $m_{air}^{dry}$ are the molecular masses of $H_2O$ and dry air respectively. Unlike TCCON $X_{CO}$ data, there are no scaling nor airmass dependent corrections for NDACC data.

The NDACC $X_{CO}$ data is calculated by the ratio between the total column of CO and the total column of the dry air. Zhou et al. (2018) pointed out that the uncertainty of the total column of the dry air is within 0.1% by using the surface pressure and NCEP water vapor. Therefore, the uncertainty of the NDACC $X_{CO}$ data is dominated by the uncertainty of the retrieved total column of CO. The uncertainty of NDACC CO total column data can be variable, depending on site-specific conditions, e.g. humidity, instrument, location and retrieval software (see Table 2). To understand the error budget for NDACC CO data, the

different contributions to the total uncertainty budget at St Denis are listed in Table 3. The systematic uncertainty is mainly coming from the spectroscopic parameters and temperature profile, while the random uncertainty is mainly coming from the SZA and temperature. Note that the systematic and random smoothing errors are not included in the reported NDACC data.

**Table 2.** The systematic and random uncertainties of NDACC retrieved CO total column.

| Site | Ny-Ålesund | Bremen | Izaña | St Denis | Wollongong | Lauder |
|---|---|---|---|---|---|---|
| sys/ran [%] | 4.0/5.0 | 3.4/4.0 | 2.1/0.5 | 2.5/1.0 | 2.1/2.2 | 2.1/1.8 |

**Table 3.** The systematic and random uncertainties for NDACC retrieved $X_{CO}$ at St Denis. '-' means that the uncertainty can be ignored.

| | Systematic [%] | Random [%] |
|---|---|---|
| Measurement | - | 0.1 |
| Spectroscopy | 2.0 | - |
| SZA | 0.1 | 0.7 |
| Temperature | 1.5 | 0.7 |
| Dry air column | 0.1 | 0.1 |
| Total | 2.5 | 1.0 |

## 3   TCCON and NDACC direct comparisons

Figure 1 shows the direct comparisons between TCCON and NDACC $X_{CO}$ co-located hourly means at the six sites. The

TCCON and NDACC measurements observe the same seasonal cycles of $X_{CO}$. At Northern Hemisphere (Ny-Ålesund, Bremen and Izaña), the seasonal variation of $X_{CO}$ is dominated by the OH variation (Té et al., 2016), with a low value of $X_{CO}$ in the summer (June-August) and a high value in the winter (December-February). At Southern Hemisphere (St Denis, Wollongong





and Lauder), the seasonal variation of $X_{CO}$ is dominated by the biomass burning, with a peak in September-November (Duflot et al., 2010). The correlation coefficients (R) at the six sites are between 0.96 and 0.99, indicating good agreement between TCCON and NDACC $X_{CO}$ measurements.

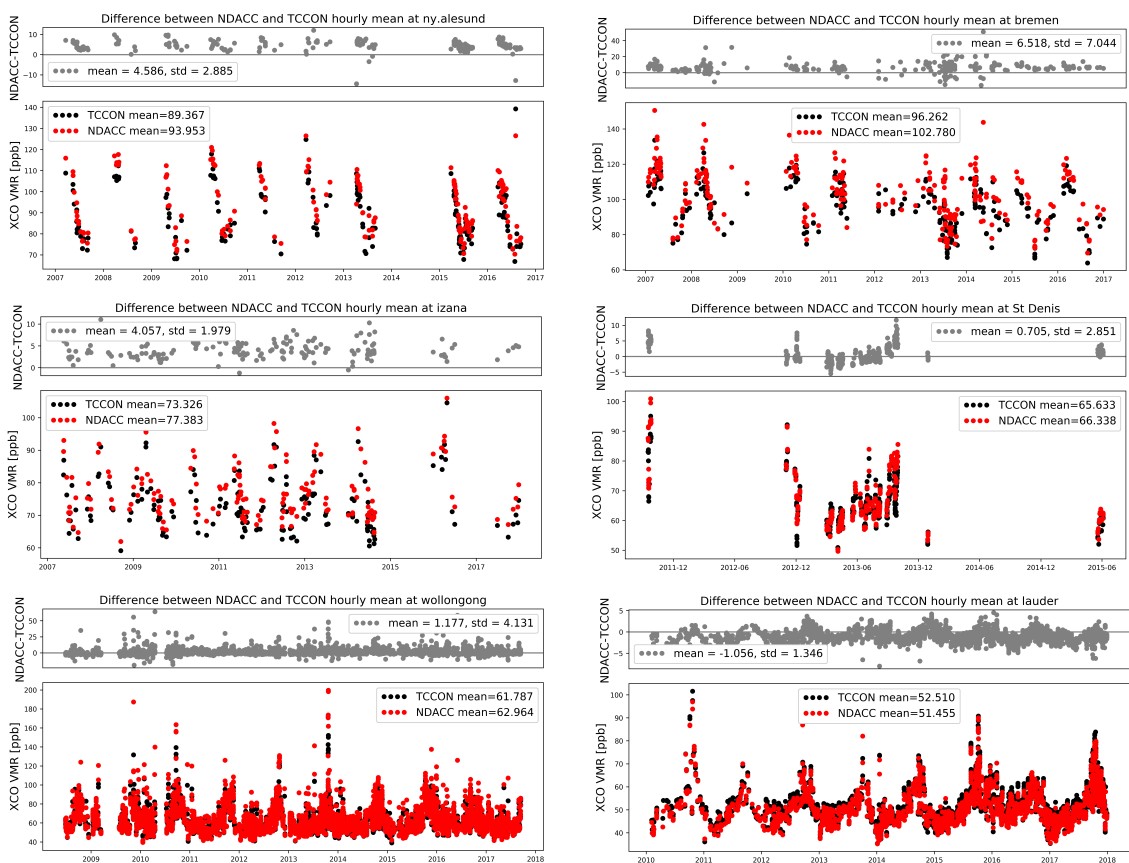

**Figure 1.** The time series of the TCCON and NDACC $X_{CO}$ measurements, together with their differences in unit of ppb. Note that the range of the y axes is different at each site due to a large variation of CO in the atmosphere.

Table 4 shows the relative mean and standard deviation (SD) between the TCCON and NDACC $X_{CO}$ measurements at these
5  sites. The mean relative biases are about 5.5% at Ny-Ålesund, Bremen and Izaña (Northern Hemisphere), and are about 0.3% at St Denis, Wollongong and Lauder (Southern Hemisphere). The difference in the mean bias between the two hemispheres is up to 5.2%. Apart from a large SD of 6.6% at Wollongong, the SDs are quite similar among other sites with a range from 2.6% to 4.3%. According to Rodgers (2003), if we ignore the smoothing error of two datasets, the systematic and random uncertainties of the differences between standard TCCON and NDACC measurements are calculated as

10  $$\varepsilon_{sys} = \varepsilon_{sys,N}, \tag{3}$$

$$\varepsilon_{ran} = \sqrt{\varepsilon_{ran,T}^2 + \varepsilon_{ran,N}^2}, \tag{4}$$





where $\varepsilon_{sys,N}$ is the systematic uncertainty of NDACC $X_{CO}$ measurements, $\varepsilon_{ran,T}$, $\varepsilon_{ran,N}$ are the random uncertainties of TC-CON and NDACC $X_{CO}$ measurements, respectively. Table 4 shows that the mean bias is higher than the systematic uncertainty at Ny-Ålesund, Bremen and Izaña, while the SD is higher than the random uncertainty at St Denis and Wollongong.

**Table 4.** The relative mean and SD between the TCCON and NDACC $X_{CO}$ measurements ((NDACC-TCCON)/NDACC×100%) at six sites, together with the systematic and random uncertainties of the differences between standard TCCON and NDACC measurements. The relative mean and SD between the TCCON and NDACC $X_{CO}$ measurements using the common optimal a priori profile. The relative mean and SD between the TCCON and NDACC $X_{CO}$ measurements using the common optimal a priori profile, but using the uncorrected TCCON data.

|  |  | Ny-Ålesund | Bremen | Izaña | St Denis | Wollongong | Lauder |
|---|---|---|---|---|---|---|---|
| Direct comparison | mean±SD [%] | 4.9±3.1 | 6.4±4.3 | 5.2±2.6 | 1.1±4.3 | 1.9±6.6 | -2.0±2.6 |
|  | sys/ran [%] | 4.0/5.8 | 3.4/5.0 | 2.1/3.0 | 2.5/3.2 | 2.1/3.7 | 2.1/3.5 |
| Common a priori profile | mean±SD [%] | 8.5±4.2 | 6.2±4.6 | 7.7±3.2 | 6.3±5.1 | 6.2±7.6 | 5.6±3.5 |
| Common a priori profile but uncorrected TCCON | mean±SD [%] | 1.5±4.2 | -0.8±4.6 | 0.7±3.2 | -0.7±5.1 | -0.8±7.6 | -1.4±3.5 |

The ground-based FTIR records the direct solar radiation, and the light path is related to the SZA. Because of the uncertainty from the spectroscopy, the TCCON $X_{CO}$ data have been corrected with an airmass dependent factor (see Eq. 1). No correction is applied to the NDACC data. To check if there is a SZA dependent in the difference between TCCON and NDACC $X_{CO}$ measurements, the differences varying with SZA are shown in Figure 2. Because of the different mean biases, the data are plotted separately in the Northern Hemisphere and in the Southern Hemisphere. In summary, the differences resulting from SZA are very small in both hemispheres, compared to the large scatter.

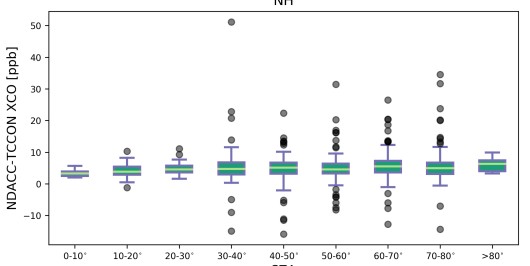
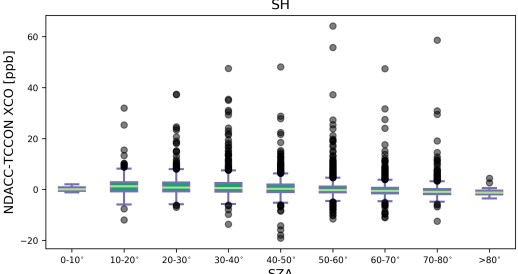

**Figure 2.** The box plot of the differences between the TCCON and NDACC $X_{CO}$ measurements as a function of SZA for Northern Hemisphere (left) and Southern Hemisphere sites (right). The bottom and upper boundaries of the box represent the 25% and 75% percentile of the data points around their a median value (green line) and the errorbars indicate the 5% and 95% percentiles of the data points.



## 4 Discussions

In this section, we investigate the difference between the TCCON and NDACC $X_{CO}$ data. Based on the optimal estimation method (Rodgers, 2000), the TCCON and NDACC retrieved $X_{CO}$ can be written as

$$X_{r,T} = \frac{TC_{r,T}}{\alpha' TC_{air}^{dry}} = \frac{1}{TC_{air}^{dry}} \left[ TC_{a,T} + \boldsymbol{A_T}(\boldsymbol{PC_t} - \boldsymbol{PC_{a,T}}) \right] + \left[ \varepsilon_{sys,T} - (1 - 1/\alpha') \frac{TC_{r,T}}{TC_{air}^{dry}} \right] \pm \varepsilon_{ran,T}, \tag{5}$$

$$X_{r,N} = \frac{TC_{r,N}}{TC_{air}^{dry}} = \frac{1}{TC_{air}^{dry}} \left[ TC_{a,N} + \boldsymbol{A_N}(\boldsymbol{PC_t} - \boldsymbol{PC_{a,N}}) \right] + \varepsilon_{sys,N} \pm \varepsilon_{ran,N}, \tag{6}$$

where, the subscript $T$ and $N$ point to TCCON and NDACC respectively, $X_r$ is the retrieved $X_{CO}$, $TC_a$ is the a priori total column of CO, $\boldsymbol{A}$ is the column average kernel, $\boldsymbol{PC_t}$ and $\boldsymbol{PC_a}$ are the true and the a priori partial column profiles respectively and $\varepsilon$ is the uncertainty. Note that, the $\varepsilon_{sys,T}$ and $\varepsilon_{ran,T}$ are the systematic and random uncertainties of the uncorrected TCCON data (without scaling correction, airmass dependent correction and using surface pressure to calculate the dry air column). $\alpha'$ represents the calculation of the dry air column and airmass independent and airmass dependent corrections in the TCCON procedure. The systematic uncertainty of the corrected TCCON data (standard product) is eliminated by its processing ($[\varepsilon_{sys,T} - (1 - 1/\alpha') \frac{TC_{r,T}}{TC_{air}^{dry}}] = 0$). It is assumed that the random uncertainty is not affected by the $\alpha'$, as $\alpha'$ is close to 1.0 and the first order of the random uncertainty is unchanged. $\alpha'$ is calculated as

$$\alpha' = \alpha \cdot \overline{TC_{O_2}/(0.2095 TC_{dry,air})} \cdot [1 + \overline{\beta \times SPF(\theta)}] = 1.076, \tag{7}$$

where $\alpha = 1.0672 \,(1\sigma : 0.0200)$ is the scaling factor in the GGG2014 code, $\overline{TC_{O_2}/(0.2095 TC_{dry,air})} = 1.016 \,(1\sigma : 0.002)$ is the difference in the dry air total column between the $O_2$ column and surface pressure, $[1 + \overline{\beta \times SPF(\theta)}] = 0.992 \,(1\sigma : 0.003)$ is the airmass dependent correction. We calculate $\overline{TC_{O_2}/(0.2095 TC_{dry,air})}$ and $[1 + \overline{\beta \times SPF(\theta)}]$ based on the TCCON measurements at these six sites.

The difference between the standard TCCON and NDACC $X_{CO}$ measurements can then be written as

$$X_{r,N} - X_{r,T} = \frac{1}{TC_{air}^{dry}} ([TC_{a,N} + \boldsymbol{A_N}(\boldsymbol{PC_t} - \boldsymbol{PC_{a,N}})] - [TC_{a,T} + \boldsymbol{A_T}(\boldsymbol{PC_t} - \boldsymbol{PC_{a,T}})]) + \varepsilon_{sys,N} \pm \sqrt{\varepsilon_{ran,N}^2 + \varepsilon_{ran,T}^2}. \tag{8}$$

Apart from the retrieval uncertainties, the difference between the TCCON and NDACC $X_{CO}$ data also includes the impact from the different a priori profiles and averaging kernels of TCCON and NDACC measurements. The a priori profile of TCCON is generated on a daily basis by the GGG2014 code (Toon and Wunch, 2014), based on MkIV balloon and ACE-FTS profiles measured in the 30-40°N latitude range from 2003 to 2007 and taking into account the tropopause height variation and the secular trend. The mean of the monthly means during 1980-2020 from the Whole Atmosphere Community Climate Model (WACCM) version 6 is used as the a priori profile for the NDACC retrievals (constant in time) at Ny-Ålesund, Bremen, Izaña, St Denis and Wollongong. The a profile profile for NDACC retrievals at Lauder is constructed from several Atmospheric Trace Molecule Spectroscopy (ATMOS) and aircraft observations. The CO a priori profiles of TCCON and NDACC measurements at these six sites are shown in Figure 3. The TCCON and NDACC a priori profiles are very different. The TCCON a priori





profiles at the six sites are close to each other in the stratosphere, which is due to the fact that the stratospheric part of TCCON a priori profile is mainly generated based on the MkIV balloon and ACE-FTS profiles measured in the 30-40°N latitude range. The TCCON a priori profiles in the troposphere at Ny-Ålesund, Bremen and Izaña are close to each other, and are very different with those at St Denis, Wollongong and Lauder. The NDACC CO a priori profiles are much more variable than TCCON a priori

profiles both in the troposphere and in the stratosphere. Based on previous studies and emission inventories, the a priori profile shapes from NDACC seem to be more realistic. For example, at St Denis, the CO VMR in the middle and upper troposphere is much larger than that in the lower troposphere, because the air in the lower altitude is pretty clean coming mainly from the Indian Ocean, while the air mass in the middle and upper troposphere is more polluted coming mainly from Africa and South America (Duflot et al., 2010; Zhou et al., 2018). The CO VMR in the boundary layer is much larger than the CO VMR in the

free troposphere at Bremen, because there is a strong anthropogenic emission at Bremen (European Commission, 2013).

The column averaging kernels (AVKs) of TCCON and NDACC retrievals are different due to their different retrieval windows, spectral resolution and retrieval settings. The AVKs of TCCON and NDACC retrievals at St Denis are shown in Figure 4. In general, the TCCON column AVK increases with altitude which implies that TCCON retrieved CO total column tends to underestimate a deviation from the a priori profile in the troposphere and to overestimate a deviation from the a priori profile

in the stratosphere. NDACC exhibits uniform sensitivity in the troposphere and varies in the stratosphere with SZA. As a result, NDACC retrieved CO total columns capture correctly a deviation from the a priori partial column in the troposphere and generally underestimate a deviation from the a priori partial column in the stratosphere.

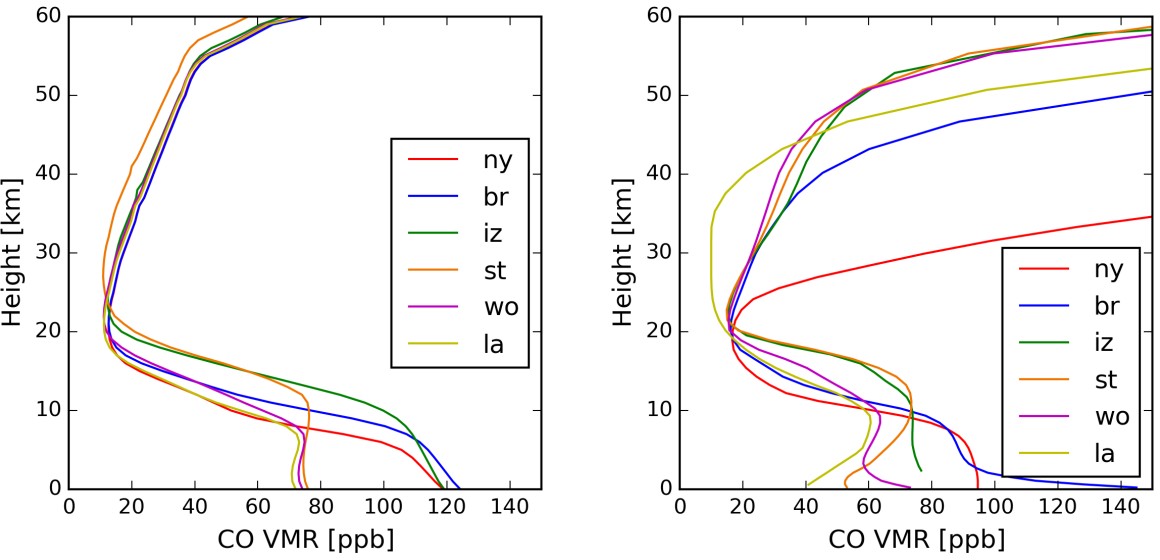

**Figure 3.** The CO a priori VMR profiles for TCCON (left) and NDACC (right) at six sites (ny: Ny-Ålesund; br: Bremen; iz: Izaña; st: St Denis; wo: Wollongong; la: Lauder). As TCCON a priori profile changes every day, the mean profiles in 2013 are shown here.



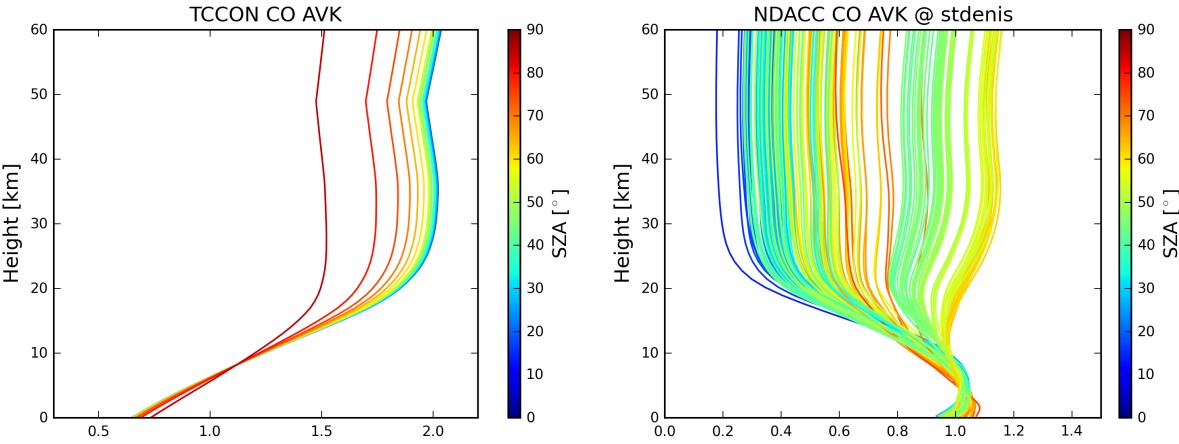

**Figure 4.** The column averaging kernels of TCCON (left) and NDACC (right) CO retrievals at St Denis.

To better compare the TCCON and NDACC retrievals, a common optimal a priori profile (subscript $op$) is applied to both TCCON and NDACC retrievals (Rodgers, 2003). The TCCON and NDACC retrieved $X_{CO}$ are

$$X'_{r,T} = \frac{1}{TC_{air}^{dry}}[TC_{op} + \boldsymbol{A_T}(\boldsymbol{PC_t} - \boldsymbol{PC_{op}})] + \left[\varepsilon_{sys,T} - (1 - 1/\alpha')\frac{TC'_{r,T}}{TC_{air}^{dry}}\right] \pm \varepsilon_{ran,T}, \tag{9}$$

$$X'_{r,N} = \frac{1}{TC_{air}^{dry}}[TC_{op} + \boldsymbol{A_N}(\boldsymbol{PC_t} - \boldsymbol{PC_{op}})] + \varepsilon_{sys,N} \pm \varepsilon_{ran,N}, \tag{10}$$

where $\boldsymbol{PC_{op}}$ is the common a priori partial column profile, $TC_{op}$ is the a priori total column, and $TC'_{r,T}$ is the uncorrected retrieved TCCON CO total column with the optimal a priori profile. The difference between the TCCON and NDACC $X_{CO}$ becomes

$$X'_{r,N} - X'_{r,T} = \frac{(\boldsymbol{A_N} - \boldsymbol{A_T}) \cdot (\boldsymbol{PC_t} - \boldsymbol{PC_{op}})}{TC_{air,dry}} + \left[((1 - 1/\alpha')\frac{TC'_{r,T}}{TC_{air}^{dry}} - \varepsilon_{sys,T}) \pm \varepsilon_{sys,N}\right] \pm \sqrt{\varepsilon_{ran,N}^2 + \varepsilon_{ran,T}^2}. \tag{11}$$

We keep the systematic uncertainty here, in case the correction of the TCCON data does not get rid of the systematic uncertainty
completely. If the optimal common a priori profile is close to the true status, then the first item in the right-band side of the Eq. 11 can be neglected and the difference between the TCCON and NDACC $X_{CO}$ data becomes

$$X'_{r,N} - X'_{r,T} \approx [((1 - 1/\alpha')X_{op} - \varepsilon_{sys,T}) \pm \varepsilon_{sys,N}] \pm \sqrt{\varepsilon_{ran,N}^2 + \varepsilon_{ran,T}^2}, \tag{12}$$

where $(1 - 1/\alpha') = 0.070$ and $X_{op} = TC_{op}/TC_{air}^{dry}$. There is a systematic (constant) difference between the TCCON and NDACC $X_{CO}$ products of about 7.0%, because of the airmass correction, airmass independent correction and the method of
calculating dry air column of TCCON data.





## 4.1 Using common a priori profile

In this section, we apply a common a priori profile for both TCCON and NDACC measurements. Instead of using another model profile, which is not always available to the TCCON and NDACC data users, we have chosen scaled NDACC a priori profiles. The scaling factor is calculated as the ratio between each retrieved NDACC CO total column and a priori CO total column

5 ($x_{N,scaled} = x_{N,ap} \times TC_{N,r}/TC_{N,ap}$). As an example, Figure 5 shows the TCCON a priori and retrieved TCCON profiles, together with NDACC a priori and scaled NDACC a priori profiles along with HIPPO CO measurements at Wollongong and Lauder. By comparing against HIPPO measurements, it is found that the vertical variability in TCCON a priori profile is too small and both the TCCON and NDACC a priori profiles have systematic biases. In summary, the scaled NDACC a priori profile is the best reasonable a priori profile among them.

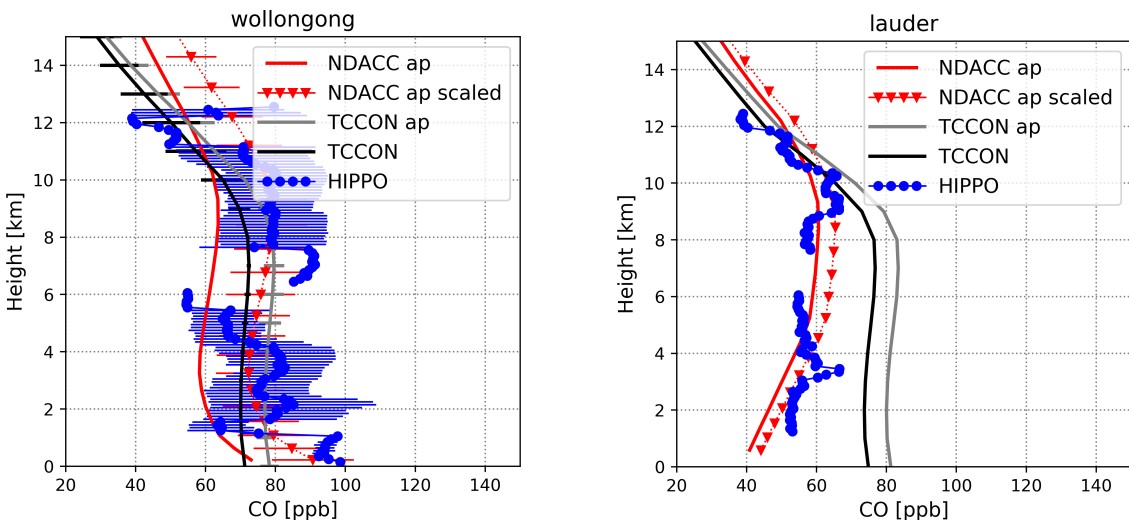

**Figure 5.** The vertical distribution of NDACC a priori profile (NDACC ap), scaled NDACC a priori profiles (NDACC ap scaled), TCCON a priori profiles (TCCON ap), TCCON retrieved profiles (TCCON) and HIPPO aircraft measurements (HIPPO) in the range from surface to 15 km at Wollongong (left) and Lauder (right). The errorbar is the SD for each dataset.

10     The systematic smoothing error is reduced by using the updated a priori profile. The differences between the TCCON and NDACC $X_{CO}$ measurements by using the scaled NDACC a priori profile as the common a priori profile are also listed in the Table 4. The biases become 5.6% to 8.5% with a mean value of 6.8%, and there is almost no inter-hemispheric dependence. However, the bias is beyond the systematic uncertainty at all sites. If we use the uncorrected TCCON data (scaling TCCON data by +7%; see Table 4), then the differences between the TCCON and NDACC $X_{CO}$ measurements at these sites become

15 -1.4 - 1.5%. It seems that the processing and correction of the TCCON data, especially the scaling factor, leads into the bias, which is consistent with the result of Kiel et al. (2016).





## 4.2 Smoothing error estimation

Although the scaled NDACC a priori profile seems to be a good candidate to represent the atmospheric CO profile, it is not the true status. According to the (Rodgers, 2003), the smoothing error should be taken into account when comparing two remote sensing retrievals

$$\sigma_s^2(TC'_{r,N} - TC'_{r,T}) = (\boldsymbol{A_N} - \boldsymbol{A_T})^T \boldsymbol{PC}_{air}^{dryT} \mathbf{S_x} \boldsymbol{PC}_{air}^{dry} (\boldsymbol{A_N} - \boldsymbol{A_T}),$$  (13)

where $\boldsymbol{PC}_{air}^{dry}$ is the partial column profile of the dry air and $\mathbf{S_x}$ is the a priori covariance estimation of the CO VMR profile in unit of $ppb^2$, including systematic and random parts. Since the scaling factor of the NDACC a priori profile is based on the NDACC retrieved total column, it is assumed that the systematic bias for the diagonal values are 2.0% (see Table 2), and then the non-diagonal elements are calculated from the diagonal values $S_{ij} = \sigma_i \sigma_j$ (von Clarmann, 2014). The random part is set as the covariance matrix of the scaled NDACC a priori profiles after smoothing with a correction width of 2.0 km. As an example, the covariance matrix at Bremen is shown in Figure 6. The random covariance is about 10 times larger than the systematic covariance. Table 5 lists the smoothing error when comparing TCCON with NDACC data by using the scaled NDACC a priori profile as the common a priori profile. The systematic smoothing error is within 0.2%, which is relatively small compared to the mean difference between the TCCON and NDACC $X_{CO}$ data (5.6 - 8.5%). The random smoothing error is between 2.0% and 4.2%, which can help to explain the large SD values in the TCCON and NDACC differences. Note that the smoothing error might be underestimated, because the CO profile in the real atmosphere does not always follow the vertical shape of the NDACC a priori profile so that the variability of CO can be larger than what we estimated.

The smoothing errors of the standard TCCON and NDACC CO total column are estimated as

$$\sigma_s^2(TC_{r,T}) = (\boldsymbol{I} - \boldsymbol{A_T})^T \boldsymbol{PC}_{air}^{dryT} \mathbf{S_{x,T}} \boldsymbol{PC}_{air}^{dry} (\boldsymbol{I} - \boldsymbol{A_T}),$$  (14)

$$\sigma_s^2(TC_{r,N}) = (\boldsymbol{I} - \boldsymbol{A_N})^T \boldsymbol{PC}_{air}^{dryT} \mathbf{S_{x,N}} \boldsymbol{PC}_{air}^{dry} (\boldsymbol{I} - \boldsymbol{A_N}),$$  (15)

where the systematic and random covariance matrices $\mathbf{S_{x,T(N)}}$ are calculated from the differences between the scaled NDACC a priori profiles and TCCON (NDACC) original a priori profiles. Table 5 shows that the systematic smoothing error of the TCCON $X_{CO}$ data can reach up to 7.9% (Lauder), which is quite large compared to the difference between TCCON and NDACC $X_{CO}$ measurement. The systematic smoothing error of TCCON data at Southern Hemisphere sites is larger than that at Northern Hemisphere sites. The random smoothing error of TCCON data is in the range between 2.0% and 3.6%, which is larger than 1.0% estimated in Wunch et al. (2015). The systematic smoothing error of NDACC data is in the range between 0.1% and 0.8% and the random smoothing error of NDACC data is about 0.3%. The smoothing error of the TCCON data is much larger than that of the NDACC data, because 1) the TCCON AVK deviates more from 1.0 than the NDACC AVK, and 2) the deviation between the TCCON a priori profile and the true atmosphere seems to be larger than that for NDACC, especially in the Southern Hemisphere.





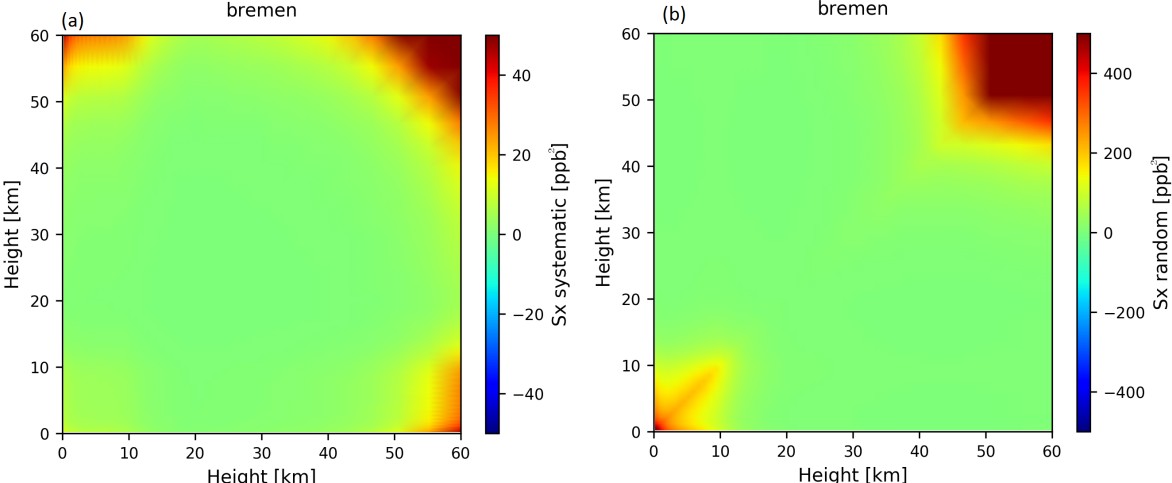

**Figure 6.** The systematic (a) and random (b) covariance matrices of the common optimal a priori profile (scaled NDACC a priori profiles) at Bremen.

**Table 5.** The systematic and random smoothing errors of the difference between TCCON and NDACC $X_{CO}$ data (using scaled NDACC a priori profiles as the common a priori profile), standard TCCON $X_{CO}$ data and NDACC $X_{CO}$ data.

| Site | Ny-Ålesund | Bremen | Izaña | St Denis | Wollongong | Lauder |
|------|-----------|--------|-------|----------|-----------|--------|
| $\sigma_s$ sys/ran [%] | 0.1/2.0 | 0.1/2.4 | 0.1/2.8 | 0.2/2.5 | 0.1/4.2 | 0.1/2.2 |
| TCCON $\sigma_s$ sys/ran [%] | 3.7/2.0 | 0.2/2.3 | 3.0/1.9 | 5.0/2.1 | 3.9/3.6 | 7.9/2.0 |
| NDACC $\sigma_s$ sys/ran [%] | 0.8/0.3 | 0.3/0.4 | 0.4/0.1 | 0.2/0.4 | 0.1/0.5 | 0.1/0.2 |

## 4.3 Comparison between AirCore and TCCON data

It is found that the difference between the TCCON and NDACC measurements with the common optimal a priori profile is higher than their uncertainties even after taking the smoothing error into account. To investigate the scaling factor (1.0672) of the TCCON $X_{CO}$ data, the AirCore measurements at Sodankylä and Orleans are compared with the TCCON $X_{CO}$ measure-
5  ments. The AirCore measurements are performed regularly by the Finnish Meteorological Institute (FMI) and the University of Groningen (RUG) at Sodankylä (Finland) since September 2013, and by the Laboratoire des Sciences du Climat et de l'Environnement (LSCE) at Orleans (France) since October 2016. Orleans and Sodankylä are operational TCCON sites but there are no NDACC $X_{CO}$ measurements available at these two sites. The AirCore measurement technique uses a balloon to bring a long winded tube up to the lower or middle stratosphere and samples a vertical profile of air inside the tube during its
10  descent. After its landing, the tube is recovered and the air inside the tube is pushed out into a gas analyser to measure the CO





mole fraction vertical profile (Karion et al., 2010). The AirCore measurements covers the vertical range from several hundred meters above the surface to about 20-25 km, and the total uncertainty of the CO measurement is about 2-3 ppb ($\sim 3.0\%$).

To compare the AirCore profiles with the TCCON $X_{CO}$ data, the AirCore profile first needs to be extended to the whole atmosphere. We use the surface in situ measurements (Schmidt et al., 2014; Kilkki et al., 2015) to fill the gap between the

surface and the lowest AirCore altitude (several hundred meters above the ground), and the scaled NDACC a priori profile to fill the CO profile above the AirCore altitude to the top of the atmosphere. It is assumed that the uncertainties are 3.0% for the surface in situ and AirCore measurements, and 6.0% for the altitude above the AirCore maximum measurement height. Second, the "extended" AirCore VMR profile is re-gridded on the TCCON retrieval levels and the partial column profile is calculated based on the surface pressure and NCEP pressure, temperature and water vapor profiles. As an example, Figure 7 shows the

"extended" Aircore profile together with the TCCON a priori profile, original AirCore and surface in situ measurements on 15 July 2014 at Sodankylä. Finally, the "extended" AirCore partial column profile is smoothed with TCCON AVK, and the $X_{CO}$ is derived from the smoothed AirCore total column

$$TC_{aircore} = TC_{a,T} + \boldsymbol{A_T}(\boldsymbol{PC_{aircore}} - \boldsymbol{PC_{a,T}}), \tag{16}$$

$$X_{aircore} = TC_{aircore}/TC_{air}^{dry}. \tag{17}$$

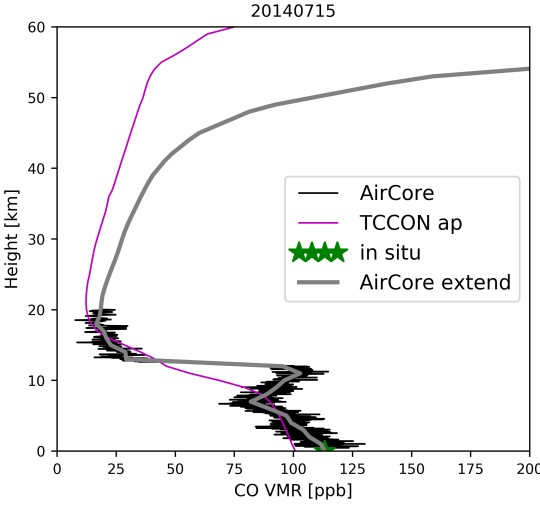

**Figure 7.** The "extended" Aircore CO profile together with the TCCON a priori profile, original AirCore and surface in situ measurements on 15 July 2014 at Sodankylä.

The co-located daily mean of the TCCON $X_{CO}$ retrievals is compared with each AirCore measurement. Instead of using 3.0% as the random uncertainty of the TCCON data, the daily SD of the TCCON data is used to represent the random uncertainty of the TCCON data. The scatter plots between the TCCON and AirCore measurements at Orleans and Sodankylä are





shown in Figure 8. The TCCON $X_{CO}$ measurements are $6.0 \pm 1.9\%$ and $6.9 \pm 2.5\%$ less than the AirCore measurements at Orleans and Sodankylä respectively. The relative differences between the TCCON and AirCore measurements have no obvious seasonal dependence. This result is consistent with Table 4 showing that the mean NDACC data is 6.8% larger than the TCCON data by using the common optimal a priori profile. Without the scaling factor (or $\alpha = 1.0000$ instead of 1.0672), the mean differences between TCCON and AirCore are -0.7% and 0.2% at Orleans and Sodankylä respectively. Further investigations are needed to understand whether the TCCON $X_{CO}$ data are incorrectly scaled at other TCCON sites.

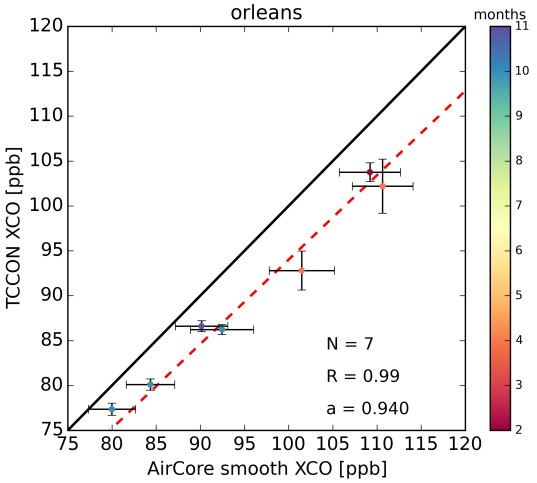
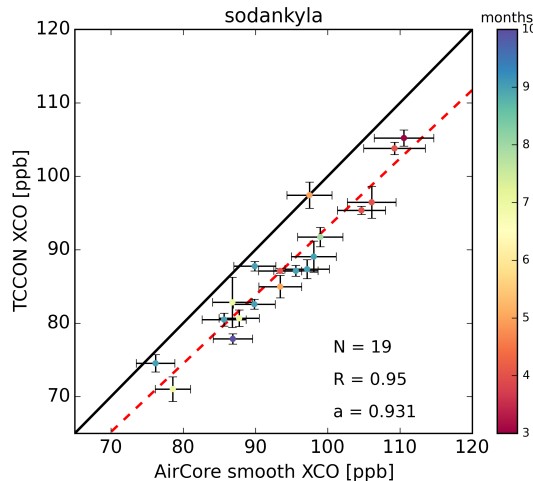

**Figure 8.** The scatter plots between the TCCON $X_{CO}$ retrievals and the smoothed AirCore $X_{CO}$ measurements at Orleans (left) and Sodankylä (right). The black line is the one-to-one line, and the red dashed line is the linear fitting (forced to cross the zero). The data is colored with the measurement month. The errorbar of the TCCON $X_{CO}$ retrieval is the daily SD, representing the random uncertainty of the TCCON data, while the errorbar of the AirCore data is the total uncertainty for each measurement. N is the number of co-located measurements, R is the correlation coefficient and a is the slope of the fitting line.

## 5  An application example

In this section, we give an example of using the TCCON and NDACC $X_{CO}$ data together to compare against an atmospheric model simulation. The TCCON and NDACC measurements from the six sites are used to compare with the Copernicus Atmosphere Monitoring Service (CAMS) operational (o-suite) reactive gas model re-analysis simulations from March 2015 to December 2018. Because there are no NDACC measurements at St Denis after June 2015, the measurements at Maïdo are used here, which is about 20 km away from St Denis (Zhou et al., 2016). The model uses the chemistry-coupled integrated forecasting system (CIFS) model run with a truncation of T511 which is approximate resolution of 40 km by 40 km and 60 vertical layers (surface to 0.1 hPa). The CAMS o-suite re-analysis CO data have been assimilated with IASI-A, IASI-B and MOPITT satellite measurements (Inness et al., 2015). The module output has a 6-hours temporal resolution. Note that the



CAMS o-suite model mainly focuses on the troposphere, and the CO VMR in the stratosphere is underestimated. More information can be found in the CAMS near-real-time system description (https://confluence.ecmwf.int/display/COPSRV/Global+production+log+files, last access: 26 April 2019) and the validation report (Wagner et al., 2019).

For each FTIR measurement, the interpolated CAMS model output is selected as one data pair, and an altitude correction is applied to the model output to make the model surface altitude to the same level of the FTIR site (Langerock et al., 2015). The time series of $X_{CO}$ from the FTIR measurements, the CAMS model with and without smoothed with the FTIR data, together with their differences are shown in Figure 9. In general, the model simulates the seasonal variation of $X_{CO}$ very well. However, the model simulation is larger than the FTIR measurements in local winter and smaller than the FTIR measurements in summer at Ny-Ålesund, indicating an underestimation in the amplitude of the seasonal variation of $X_{CO}$ for the CAMS model. Several high $X_{CO}$ FTIR measurements are not well captured by the CAMS model at Ny-Ålesund and Bremen. Both TCCON and NDACC measurements show many high $X_{CO}$ values at Wollongong, which are not well simulated in the CAMS model. There is an extremely high value in the CAMS model simulations at Lauder, which is not observed in TCCON and NDACC measurements.

Table 6 lists the mean and SD of the relative difference between the CAMS model (with and without smoothing) and FTIR measurements. The averaged bias between the TCCON and CAMS smoothed data is 5.2%, while the averaged bias between the NDACC and CAMS smoothed data is -1.2%. The latter bias is due to the underestimation of the stratospheric CO in the CAMS model. The difference between the averaged biases of the CAMS model with TCCON and NDACC data is 6.4%, which is consistent with the result obtained when comparing TCCON and NDACC $X_{CO}$ data using the scaled NDACC a priori profile as the common a priori profile (see Table 4). According to the AirCore measurements, the bias of 5.2% between the TCCON and CAMS smoothed data is mainly due to the scaling factor of the TCCON $X_{CO}$ measurements. In addition, Table 6 shows that the changing of the model $X_{CO}$ data after smoothing with TCCON data ranges from 2.1 to 6.1%, which is much larger than that after smoothing with NDACC data of 0.3 - 2.4%. It is confirmed that the smoothing error of TCCON $X_{CO}$ data is much larger than that of NDACC $X_{CO}$ data, and the smoothing error must be taken into account when using FTIR $X_{CO}$ data.

**Table 6.** The mean and SD of the relative difference between the CAMS and FTIR (TCCON and NDACC) $X_{CO}$ data, with and without smoothing. St Denis*: TCCON data is from St Denis site, while NDACC data is from Maïdo site.

| (CAMS-FTIR)/FTIR [%] | TCCON | TCCON smooth | NDACC | NDACC smooth |
|---|---|---|---|---|
| Ny-Ålesund | 3.4±5.5 | 7.6±6.0 | 1.1±6.1 | -1.3±6.1 |
| Bremen | 1.4±6.0 | 3.5±6.0 | -1.6±5.8 | -3.5±5.4 |
| Izaña | 2.1±5.2 | 5.2±5.2 | -3.1±4.2 | -3.6±4.2 |
| St Denis* | -1.0±5.1 | 4.7±4.1 | -0.0±4.0 | -0.8±4.0 |
| Wollongong | -2.3±6.8 | 2.1±6.8 | -2.8±9.2 | -3.1±9.2 |
| Lauder | 2.0±10.9 | 8.1±8.1 | 5.3±7.7 | 4.3±7.0 |





**Figure 9.** The time series of $X_{CO}$ from the TCCON measurements, the CAMS model and the CAMS model smoothed with TCCON data at six sites (first column) and their relative differences (second column). The time series of $X_{CO}$ from the NDACC measurements, the CAMS model and the CAMS model smoothed with NDACC data at six sites (third column) and their relative differences (last column).

## 6 Conclusions

In this study, the difference between the TCCON and NDACC $X_{CO}$ data products during the period 2007-2017 has been studied at six sites (Ny-Ålesund, Bremen, Izaña, St Denis, Wollongong and Lauder) where co-located NDACC and TCCON FTIR observations are carried out.





When doing a straightforward comparison between both $X_{CO}$ data products, it is found that for the Northern Hemisphere sites the TCCON $X_{CO}$ values are about 5.5% smaller than the NDACC $X_{CO}$ values while they agree well in the Southern Hemisphere. To understand these inter-hemispheric differences in the biases, we have looked into more detail in the characteristics of both products, in particular their averaging kernels and dependence on the a priori profiles used in the retrievals.

Taking into account these differences in the comparisons, by adjusting the products towards a common optimal a priori profile, it is found that the biases between the adjusted TCCON and NDACC $X_{CO}$ data product are almost constant (5.6 - 8.6%) with a mean value of 6.8%; for the common optimal a priori profile we have chosen the NDACC a priori profiles scaled with the ratios of the retrieved to the a priori columns.

The first conclusion therefore is that the apparent inter-hemispheric difference in the bias disappears when accounting cor-

rectly for the smoothing errors. To confirm this first finding we have estimated the systematic and random smoothing errors of the TCCON and NDACC $X_{CO}$ data according to the optimal estimation method (Rodgers, 2000): the TCCON $X_{CO}$ systematic smoothing errors vary in the range between 0.2% (Bremen) and 7.9% (Lauder), and their random smoothing errors lie in the range between 2.0% and 3.6%, which is larger than the random uncertainty of 1.0% estimated in Wunch et al. (2015). Also the TCCON $X_{CO}$ systematic and random smoothing errors are larger than the NDACC $X_{CO}$ systematic and random smooth-

ing errors that are in the range between 0.1% and 0.8% for the systematic ones and of order 0.3% for the random ones, and they are larger in the Southern than in the Northern hemisphere. This is because 1) the TCCON AVK deviates more from 1.0 than the NDACC AVK, and 2) the deviation between the TCCON a priori profile and the true profile seems to be larger than that for NDACC, especially in the Southern Hemisphere. This finding also demonstrates the importance of accounting for the smoothing errors when comparing FTIR $X_{CO}$ data, and particularly TCCON $X_{CO}$ data, with satellite measurements or model

simulations. This has not always been done in recent satellite validation studies (Borsdorff et al., 2016, 2018; Hochstaffl et al., 2018). As a consequence, the biases reported in these papers are not relevant because they fall in the systematic uncertainty, especially in the Southern Hemisphere.

Our second conclusion is that the remaining 6.8% bias between the TCCON and NDACC $X_{CO}$ data (when using the common optimal a priori profile) originates in the scaling correction that has been applied to the standard TCCON data. To

demonstrate this second finding we have compared AirCore in situ profile measurements with the standard TCCON $X_{CO}$ data. It is found that the TCCON $X_{CO}$ measurements are $6.0 \pm 1.9\%$ and $6.9 \pm 2.5\%$ smaller than the AirCore measurements at Orleans and Sodankylä, respectively, which is consistent with the bias found between the TCCON and NDACC $X_{CO}$ measurements. Eliminating the scaling correction (setting $\alpha = 1.0000$ instead of 1.067), the differences between the TCCON and AirCore measurements become -0.7% and 0.2% at Orleans and Sodankylä, respectively. A similar confirmation is found

when comparing the TCCON $X_{CO}$ data to CAMS assimilation analyses. Further investigations should therefore be carried out in the TCCON community to study the CO scaling factor based on comparisons with in situ CO profile observations (e.g. calibrated aircraft or AirCore measurements) at additional TCCON sites.



*Data availability.* The TCCON GGG2014 data are publicly available through the TCCON database (https://tccondata.org/). The details of the TCCON data for each site, please refer to Notholt et al. (2014b, a); Blumenstock et al. (2014); De Mazière et al. (2014); Griffith et al. (2014); Sherlock et al. (2014); Warneke et al. (2014); Kivi et al. (2014); Kivi and Heikkinen (2016). The NDACC data are publicly available from the NDACC website (http://www.ndacc.org).

*Competing interests.* The authors declare that they have no conflict of interest.

*Acknowledgements.* This study is supported by the EU-funded C3S_311a_Lot3 project. The TCCON site at Reunion Island is operated by the Royal Belgian Institute for Space Aeronomy with financial support in 2014, 2015, and 2016, 2017 under the EU project ICOS-Inwire and the ministerial decree for ICOS (FR/35/IC2) and local activities supported by LACy/UMR8105 - Université de La Réunion. The NDACC and TCCON stations Ny-Ålesund, Bremen, Izaña have been supported by the German Bundesministerium für Wirtschaft und Energie (BMWi)

via DLR under grants 50EE1711A-B. The Lauder FTIR measurements are core funded by NIWA from New Zealand's Ministry of Business, Innovation and Employment through the Strategic Science Investment Fund. We thank the HIPPO team for making the aircraft measurements available http://hippo.ucar.edu/.

*Author contributions.* MZ, CV and MDM designed the study. MZ wrote the paper and produced the main analysis and results with significant input from BL. HC and MR, RK provide the AirCore measurements. MKS, CH, JMM, PH, DS, DFP, NJ, VAV, OEG, MS, MP, TW provided

and analyzed the TCCON and NDACC measurements. All authors read and provided comments on the paper.





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
