# Peer review of "TCCON and NDACC $X_{CO}$ measurements: difference, discussion and application"

_Atmospheric Measurement Techniques, 2019_

## Referee Comment (RC1) · Anonymous Referee #1 · 31 Aug 2019

In this paper, the TCCON and NDACC XCO measurements are compared against each other on six sites. The methods of calculating the XCO for TCCON and NDACC are discussed, and the differences between two datasets have discussed and investigated. The bias in XCO between TCCON and NDACC is about 5.5% at three Northern Hemisphere sites, but it is about 0.3% at three Southern Hemisphere sites. The hemispheric dependence in bias is attributed to their smoothing errors. The smoothing error of TCCON data is relatively large compared to NDACC data, resulting from its averaging kernel and a priori profile choice. After using the scaled WACCM model data as the a priori profiles for TCCON and NDACC measurements, the biases at six sites become relatively consistent (5.6-8.5%). The remaining ~7% in bias is mainly due to the scaling factor of the TCCON data. The uncertainties of both datasets are discussed

in the paper. By comparing with AirCore measurements, the TCCON data is 6-7% underestimated. Meanwhile, the error in the fitting slope is about 2%, which is less than the bias. This bias in the TCCON XCO data should be corrected in the TCCON community as the TCCON data is widely applied for satellite and model validations. At last, the authors show an example of comparing TCCON and NDACC measurements with CAMS model data. This highlights that the smoothing correction must be applied when comparing with FTIR XCO data, especially for TCCON. Meanwhile, it also shows that the TCCON XCO is about 5.2% larger than the CAMS data, which has been assimilated with IASI-A –B and MOPITT satellite observations. The paper is nicely written with illustrative figures and I don't see any obvious errors. The paper is important for the TCCON community, as a systematic bias is found in their XCO data. Data users should be pay attention to consider the smoothing correction when comparing the ground-based FTIR measurements with satellite observations or model simulations. I therefore suggest it can be published after correcting/considering a few relatively minor points.

P4 line 10-11. In this paper, the 3% is adopted as the TCCON uncertainty for all six sites, while NDACC data has different uncertainties at different sites?

P4 line 17. "0.0035 - 0.007" to "0.0035 - 0.0070"

P8 line 23. Please write the full name for "MkIV" and "ACE-FTS" for the first time

Table 5. Why the TCCON systematic smoothing error at Bremen is only 0.2%, which is much less than other sites?

[Figure]

---

## Referee Comment (RC2) · Anonymous Referee #2 · 3 Sep 2019

Zhou et al. provide a comprehensive intercomparison between two remote sensing Fourier transform infrared (FTIR) spectrometer measurement techniques: sensing carbon monoxide (CO) in the near infrared (NIR) and mid-infrared (MIR) spectrum. They compare column average CO volume mixing ratios (X$_{CO}$) at 6 sites that are a part of both the TCCON and NDACC networks. The study makes a direct comparison, as well as analyzes the impact of the uncertainty budget on the difference found between the two methods. Overall, this study presents a valuable comparison that allows for future synergy when using CO measurements from both the TCCON and NDACC networks.

The subject matter of measurement intercomparison is well within the scope of AMT. The paper meticulously describes the methodology, and draws on previous work in the field. Writing style is clear and descriptive. The study includes rigorous analysis

and sound mathematical procedures. Subsequent to addressing and/or responding to the minor comments and clarifications below, I recommend that the manuscript be accepted for publication in AMT.

**Specific Comments:**

*Comment 1: AirCore comparison*

Using the NDACC a priori profile in section 4.3 to extend the AirCore profile presents a circular argument. It is not necessarily surprising that the AirCore total column was higher than the TCCON total column, because the scaled NDACC a priori profile was used to "fill in the blanks".

(a) First of all, it is unclear where the "scaled NDACC a priori" (P14, L5) originates, because it is stated that Sodankylä is not an operational NDACC site (P13, L8). Please clarify how the scaled a priori is calculated - i.e. are there non-operational data available? If so, are they up to the same standard as the rest of the NDACC network?

(b) I suggest to use the TCCON a priori above the top of the AirCore measurements, perhaps scaled by the AirCore:TCCON profile difference that exists between 15 to 20 km. Figure 5 suggests that the shape of the TCCON a priori profile above 10 km might be better than the NDACC at Lauder and Wollongong. While the profile above 20 km may not contribute a large amount to the total column, it is important to remove potential bias in the comparison. Alternatively, the authors could calculate how much the "extended profile" contributes to the total column and explain why it doesn't convolute the results.

*Comment 2: "Low" Mean Bias for the Southern Hemisphere*

It is stated that the Southern Hemisphere has a low mean bias of 0.3% (P6, L6) and that the values "agree well" (P18, L2). I am a little nervous about how good the 0.3% seems, because it is the result of averaging positive and negative numbers. In particular, the Lauder station uses different a priori protocol than Wollongong and St Denis (P8, L25-28). If only the two latter stations are included, the Southern Hemisphere bias is 1.5%, which may be more representative. I would appreciate a small discussion of this in the

manuscript. Perhaps it would also be useful to discuss the absolute relative error.

*Comment 3: Double-check main text values with figures and tables for consistency*
(a) Figure 1 - Bremen shows a mean difference of 6.518 +/- 7.044 ppb. This seems inconsistent with Table 4 Bremen 6.4 +/- 4.3%. (b) P16, L21-22 - the values in the data range do not match Table 6. For example, I think 2.1 to 6.1% should be 2.1 to 8.1%.

*Comment 4: Relevance of calculations on Page 10*
It was unclear to me why the systematic uncertainty was kept here (Eq. 12) while on P8, L12 it was eliminated. Also, how is the choice of optimal common a priori made to ignore the first component (P10, L10)? The comment on P12, L2-3 suggests that the first component should not be ignored. There is a missing link/clarification needed between the theory presented on this page and exactly how it relates to this study. Also, would the main text on page 10 work better inside section 4.1?

**Minor Comments:**
*P1, L10-11:* Reword to clarify what is meant by this sentence. For example, does this convey what is meant: "The TCCON smoothing error is significant because it is higher than the reported uncertainty".
*P2, L21:* Mention briefly why the measurements are assimilated into models.
*P3, L29  L30:* Describe acronym ATM and JPL.
*P3, L32:* Add the equation definition after the "CO total column".
*P4, Table 1:* I suggest site reference papers rather than "Research group".
*P4, L1:* What is meant my "indirectly validated" against AirCore measurements?
*P4, L11:* Please clarify why 3.0% and not 3.5% is used as the upper limitation on random uncertainty.
*P4, L12-14:* I am confused by the last sentence in section 2.1. How does it relate to WMO scaling? Was public data not used in this study?
*P5:* I suggest to swap the order of Tables 2 and 3, and move sentence line 8-9 to the end of the paragraph.
*P5, L12:* Unclear purpose for the last sentence of Section 2.2. Are the authors

suggesting these smoothing errors be included in reported NDACC data?

*P5, Table 3 caption:* Mention that the "Total" uncertainties are calculated by adding the sub-types in quadrature. Also, why can "-" be ignored? Are they simply too small to worry about?

*P5, Fig1:* Is the seasonality in the Lauder difference plot related to that NDACC station using a different a priori than the other NDACC stations?

*P7, Table 4 caption:* I suggest to work on reducing the repetitive wording.

*P7, L4-L9:* Is the conclusion from this section that NDACC adequately accounts for SZA dependence in the retrieval algorithm?

*P11, L3:* Describe why the scaled NDACC a priori were used as common a priori, rather than the NDACC a priori.

*P11, Figure 5:* Please add uncertainty/standard deviation to the HIPPO data points.

*P11, L13-14:* Unclear why 7% (links to specific comment #4).

*P12, L8:* The 2.0% assumption on diagonal systematic bias is lower than any value presented in Table 2 - clarify why this value was chosen.

*P12, L25-26:* Discuss why the random smoothing error is larger here than in Wunch et al. (2015).

*P14, L2:* Briefly mention how diffusion inside the AirCore coil can impact the measurement, and how it is minimized.

*P14, L6-7:* How are the 3.0% and 6.0% assumed uncertainties chosen?

*P15, L13:* What is "a truncation of T511"? Does it relate to the model altitude level?

*P16, L4:* Is the CAMS model interpolated in both time and space?

*P16, L10:* Mention that fewer satellite observations improve the CAMS model at higher latitudes due to measurement difficulties, which may cause the poorer performance at Ny-Ålesund.

*P16, L13:* Mention that high locally impacted values are not expected to be captured by the model due to dilution: both temporally (6 hr compared to minutes) and spatially (40 km square compared to site location).

*P17, Figure 9:* Consider moving this to a supplement because details are in Table 6.

**Technical Corrections:**
The paper is well written and there are only minor technical corrections.

*P1, L5:* A direct comparison shows the NDACC $X_{CO}$ measurements...
*P1, L14:* To determine the source of the bias, regular...
*P2, L8:* ...), and biomass burning (...
*P2, L8:* There are also small quantities of CO...
*P2, L11:* biomass burning
*P2, L13:* ...thus affects the...
*P3, L1:* Despite the similar...
*P3, L14:* ...carried out in Section 3.
*P3, L15:* ...investigated in relation to...
*P4, L13:* ...in public TCCON...
*P4, L19:* The reference spectroscopy database is...
*P5, L15:* At Northern Hemisphere stations...
*P5, L17:* At Southern Hemisphere stations...
*P6, L1:* ...dominated by biomass burning...
*P8, L1:* ...investigate the causes of the difference between...
*P8, L27:* The a priori profile...
*P9, L7:* ...is relatively clean, coming mainly...
*P9, L9:* At Bremen, the CO VMR in...
*P9, L10:* ... free troposphere, because there are strong local anthropogenic emissions
(European Commission, 2013).
*P9, L16:* ...total columns correctly capture a...
*P9, Fig. 3 caption:* ...a priori profiles change every day,...
*P11, L9:* ...is the most reasonable...
*P13, L5:* ...measurements have been performed...

*P13, L9:* ...a long coiled tube...

*P13, L10:* ...tube is transferred to a gas analyser...

*P15, L15:* The model output...

*P16, L10:* ...model at this site.

*P16, L19:* According to the AirCore measurements in Sect. 4.3, the bias...

*P18, L6:* ...data products are more consistent (5.6...

––––––––––––––––––––––––––––––––

---

## Author Comment (AC2) · 13 Sep 2019

The comment was uploaded in the form of a supplement:
https://www.atmos-meas-tech-discuss.net/amt-2019-266/amt-2019-266-AC2-supplement.pdf